# Radiomics-Clinical AI Model with Probability Weighted Strategy for Prognosis Prediction in Non-Small Cell Lung Cancer

**DOI:** 10.3390/biomedicines11082093

**Published:** 2023-07-25

**Authors:** Fuk-Hay Tang, Yee-Wai Fong, Shing-Hei Yung, Chi-Kan Wong, Chak-Lap Tu, Ming-To Chan

**Affiliations:** School of Medical and Health Sciences, Tung Wah College, Hong Kong, China

**Keywords:** radiomics, radiotherapy, artificial intelligence (AI), prognosis prediction, clinical factors, machine learning, non-small cell lung cancer

## Abstract

In this study, we propose a radiomics clinical probability-weighted model for the prediction of prognosis for non-small cell lung cancer (NSCLC). The model combines radiomics features extracted from radiotherapy (RT) planning images with clinical factors such as age, gender, histology, and tumor stage. CT images with radiotherapy structures of 422 NSCLC patients were retrieved from The Cancer Imaging Archive (TCIA). Radiomic features were extracted from gross tumor volumes (GTVs). Five machine learning algorithms, namely decision trees (DT), random forests (RF), extreme boost (EB), support vector machine (SVM) and generalized linear model (GLM) were optimized by a voted ensemble machine learning (VEML) model. A probabilistic weighted approach is used to incorporate the uncertainty associated with both radiomic and clinical features and to generate a probabilistic risk score for each patient. The performance of the model is evaluated using a receiver operating characteristic (ROC). The Radiomic model, clinical factor model, and combined radiomic clinical probability-weighted model demonstrated good performance in predicting NSCLC survival with AUC of 0.941, 0.856 and 0.949, respectively. The combined radiomics clinical probability-weighted enhanced model achieved significantly better performance than the radiomic model in 1-year survival prediction (chi-square test, *p* < 0.05). The proposed model has the potential to improve NSCLC prognosis and facilitate personalized treatment decisions.

## 1. Introduction

Lung cancer is one of the leading commonly diagnosed cancer, accounting for 11.6% of cancer cases. It has the highest mortality among all malignancies worldwide, comprising approximately 25% of all cancer death. Non-small cell lung cancer (NSCLC) contributes to the majority of lung cancer incidence, adding up to almost 85% of cases [1]. The primary treatment modalities for NSCLC are surgery, radiation therapy and chemotherapy. Recent research indicated that patients may benefit from immunotherapy for NSCLC with specific biomarkers [2]. Moreover, targeted therapy is favorable for NSCLC with specific genes or proteins [3]. Formulating a treatment plan and patient management is crucial to the prognosis of NSCLC. Traditionally, TNM staging is a widely used system for prognosis stratification and treatment decision for NSCLC based on tumor size (‘T’), lymph node involvement (‘N’) and distant metastasis (‘M’). However, the TNM staging system only provides a stratified prognosis prediction based on the characteristics of the tumor, which is not personalized for each patient. Furthermore, other prognostic factors that are influential to the outcomes of patients, such as age and histology, are not taken into consideration by the TNM staging system. Due to the limitations of the TNM staging system, there is a need to incorporate other factors that can provide more comprehensive and individualized predictions.

Radiomics is a rapidly growing field that uses quantitative data extracted from medical images, such as computed tomography (CT), magnetic resonance imaging (MRI), and position computed tomography (PET), to provide a more detailed characterization of tumors [4]. These data that include textural and morphological information can be used to identify subtle differences in the heterogeneity of tumor that are significant to the treatment outcome [5] and personalized medicine [6].

Machine learning has been used in radiomics in predicting treatment outcomes of cancer patients in colorectal cancer [7], head and neck cancer [8], hepatocellular carcinoma [9], and NSCLC [10]. Common machine learning algorithms include decision tree (DT), random forest (RF), extreme boost (EB), support vector machine (SVM), and generalized linear model (GLM) [11].

Chaddad et al. (2017) investigated the use of radiomics in predicting the survival time of patients with NSCLC based on the shape and the textural radiomic features [12]. The subjects were classified according to their histology and TNM staging information. Twenty-four radiomic features were used. The study suggested that these radiomic features have the potential ability to predict the survival time of patients with area under the curve (AUC) from 0.70 to 0.76. Le et al. (2021) performed another study to evaluate the predictive ability of radiomics in one-year, three-year, and five-year survival of NSCLC patients. A risk score was developed from ten radiomic models with AUC of 0.696, 0.705 and 0.657 for one-year, three-year, and five-year survival, respectively [13].

Ching et al. (2023) used a combined radiomic model with clinical features (RC combined model) for prostate cancer for the prediction of five-year progression-free survival prognosis and obtained an AUC of 0.797 [14]. Their model combined radiomic factors with clinical factors using ridge regression. The best accuracy of RC combined model obtained was 0.729. Their result is still not impressive.

It appears that radiomics is helpful in the early detection of survival for NSCLC patients [15]. In this study, we present a radiomics-clinical probability weighted enhanced model for the prediction of prognosis of NSCLC. The model combines radiomic features extracted from computed tomography (CT) images with clinical factors to predict the overall survival of NSCLC patients. The model is based on a combination of machine learning algorithms that include radiomics features and clinical information using a probability weighted strategy.

## 2. Materials and Methods

### 2.1. Data Acquisition

Pre-treatment planning CT images were acquired from The Cancer Imaging Archive (TCIA). TCIA is an open-access database managed by the Frederick National Laboratory for Cancer Research. It is funded by the Cancer Imaging Program (CIP) of National Cancer Institute (NCI) in the United States [16]. The images were reviewed and approved by TCIA Advisory Group, which is formed by experts in cancer imaging, informatics, and related technology to ensure the reliability of the database. TCIA contains medical images of different types of cancer. Supporting information of the images, such as age, gender, and outcomes of the patients, are also provided if available.

Four hundred and twenty-two NSCLC patients’ information was retrieved from TCIA. All patients received radiotherapy with curative intent. The dataset contains pre-treatment planning CT images with radiotherapy structures. Gross tumor volume (GTV) was segmented manually by experienced oncologists. Patients’ demographics and tumor information, including age, gender, TNM staging, and histology, were also acquired from the database.

### 2.2. Case Selection

The samples were selected by convenience with all samples available in the target database. Among all 422 cases collected, 5 cases with distant metastasis or with GTV outside the lung were excluded from the study. A total of 8 cases were ignored due to errors in acquiring the DICOM images. A total of 57 cases with missing data in age, histology, T stage, or overall staging were also excluded. Finally, 352 cases were included. By using G*Power analysis (version 3.1.9.7, Statistical Consulting Group, UCLA, Los Angeles, CA, USA), the sample size required is 52 with a power of 0.8, effective size 0.5, and α at 0.05 for chi-square test. To give more accurate machine learning results, we used 352 samples in this study.

### 2.3. Feature Extraction

Cases with multiple GTVs were combined into a single GTV for feature extraction by the Eclipse treatment planning system, version 15.6 (Varian, Palo Alto, CA, USA). The GTV was utilized for radiomic feature extraction performed by 3D slicer (v. 5.2.1, slicer.org) with Pyradiomics extension (Computational Imaging and Bioinformatics Lab, Harvard Medical). A total of 107 radiomic features were extracted from each sample (Table 1), which were imported into the machine learning algorithms. These radiomic features can be classified into seven groups, including shape, first-order feature, gray level co-occurrence matrix (GLCM), gray level dependence matrix (GLDM), gray level run length matrix (GLRLM), gray level size zone matrix (GLSZM), and neighborhood gray-tone difference matrix (NGTDM) (see Table 1).

### 2.4. Study Endpoints

Overall survival (OS) was defined as the time from the patient having radiotherapy treatment to the time of death when the precise cause of death is not specified. Luna et al. (2022) evaluated the prediction of overall survival (OS) using radiomics on patients with stage III lung adenocarcinoma treated with chemoradiation. It revealed that by integrating radiomic features into a baseline Cox model based on age and ECOG performance status scale, there was an improvement in the OS predictive ability of the model [15]. In our study, we divided the study endpoints to one-year, three-year, and five-year OS so that we have a more precise prediction model.

To avoid overfitting and bias due to uneven data, a balanced sample with an equal sample size in each treatment outcome was randomly selected at each endpoint for validation and testing of the models (Table 2).

### 2.5. Machine Learning for Data Processing

The radiomic features extracted were imported into the machine learning algorithms using R (Ihaka and Gentleman; v. 4.1.3, Switzerland) [17] with Rattle package [18]. The machine learning algorithms used in the study include decision trees (DT), random forests (RF), extreme boost (EB), support vector machine (SVM), and generalized linear model (GLM). We built our AI model by randomly splitting the sample into three independent cohorts, with 70% of the sample in the training cohort to identify patterns, 15% of the sample in the validation cohort to measure our progress, and 15% of the sample in the testing cohort to evaluate the performance of the model on unobserved data. The predicted treatment outcome was quantified as binary classification: a score of less than 0.5 indicated the model prediction of the patient survived at the given endpoint, while a score of greater than 0.5 signified that the model predicted the patient did not survive.

The above machine learning algorithms were optimized by the voted ensemble machine learning (VEML) model we proposed earlier [18]. Due to the difference in the properties of machine learning algorithms, each algorithm has its own limitations. A study stated that VEML demonstrates an improvement in predictive performance when compared with a single machine-learning algorithm [19]. Hence, the ensemble method was introduced to compensate for the weaknesses of the different models in order to achieve a higher prediction accuracy. This method incorporates results from the five machine learning algorithms, which are decision tree (DT), random forest (RF), extreme boost (EB), support vector machine (SVM), and generalized linear (GLM), by calculating the average score of the majority predicted outcome by these algorithms that were alive or dead using ROC analysis [20]. (Figure 1).

Prediction of prognosis using radiomic model or clinical factors model have their own strengths and weaknesses. The radiomic model is a non-invasive tool that predicts cancer prognosis by mathematical analysis of radiomic features. For clinical factors model, it provides a subjective measurement based on clinical elements, such as age and histology, that may significantly influence the prediction results. On the other hand, the TNM staging system only stratifies patients according to the tumor size, lymphatic involvement, and the extent of metastasis, but not personalized for each patient [21]. Hence, a weighted method was proposed to construct a combined probability-enhanced model, which is a weighted combination of radiomic model and clinical factors model (Figure 2). By combining the two models, it can take the strengths of each model and potentially improve the accuracy of the predicted outcome.

### 2.6. Probability Weighted Enhanced Model (PWEM)

The association of patient demographics and clinical factors with radiomics features have proven to add further value to the predictive power of machine learning models [22].

A significant correlation was discovered between advanced age and AJCC TNM staging with the survival of the patients [23,24]. It appears that further consideration needs to be explored by taking advantage of patient clinical factors combined with radiomics features for machine learning data mining.

The Probability Weighted Enhanced Model (PWEM) is a multi-algorithm model proposed in this study to facilitate collaborative voting between radiomics and the clinical factor model (Figure 2). The rationale behind this is to account for crucial and high-risk clinical factors as references, in order to produce a more realistic prediction. It consists of hard-voting and soft-voting techniques for decision-making by considering the numerical outcomes of radiomics features and categorical clinical factors. The hard voting consists of performing VEML on the radiomics features model and clinical factors model separately, as a result of both radiomics and clinical factors would have a VEML score indicating the probability and prediction for the outcome. For soft voting, a classifier known as predictive weighting classifies the weighting of the radiomics model and clinical factor model based on probability.

Predictive weighting is an important factor that reflects the model’s probability of acquiring a correct prediction under a conflicting situation. When the radiomics model and the clinical factor model have different predictions of the patient outcome, the occurrences of a correct prediction by each model are counted according to the probability of getting a correct prediction by each model.

The weighted score of the PWEM Model reflects the collective survival prediction of the radiomics model and clinical factor model. It is deduced by combining the radiomics model score and a clinical model score while multiplying their corresponding predictive weighting factor. The weighted score is presented in a numerical value between 0 and 1, a value less than 0.5 indicates the PWEM model has predicted the patient to survive, while a value equal to 0.5 or larger than 0.5 indicates the PWEM model has predicted the patient to be dead (Figure 2). It is calculated by the following equation:Weighted Score = Radiomic VEML Score × Radiomics Weighting + Clinical VEML Score × Clinical Weighting

## 3. Results

### 3.1. Patient Demographics and Tumor Characteristics

A total of 352 patients with NSCLC were included in the study. The overall staging was classified according to the TNM system of the American Joint Committee on Cancer (AJCC). Among the patients, 67% were male, while 33% were female. A total of 44% of the patients were diagnosed with stage IIIB NSCLC. For histology, the highest proportion of patients were diagnosed with squamous cell carcinoma, which was equivalent to 40% of the sample (Table 3).

### 3.2. Prognosis Prediction Performance of the Models at Different Endpoints

Receiver Operating Characteristics (ROC) curves were utilized to evaluate the performance in prognosis prediction of radiomic model, clinical factors model and the combined probability-weighted enhanced model at the endpoints of one-year, three-year, and five-year survival. The area under the curve (AUC) of ROC curves at each endpoint was generated by Rattle in R.

### 3.3. Performance Analysis for Machine Learning Models

For the predictive performance for the one-year, three-year and five-year endpoints, the overall average performance of the radiomics model (RAT), clinical model (CF) and the Probability Weighted Enhanced (PWE) model obtained AUCs of 0.941, 0.856 and 0.949, respectively. The RAT model and PWE model had similar performance for survival prediction, and both the RAT and PWE model outperform the CF model (Figure 3, Figure 4 and Figure 5).

The best performance was achieved by the PWE model for the one-year survival prediction with an AUC of 0.955 (95% CI [0.9264,0.9742]); with the RAT model for the five-year survival prediction with an AUC of 0.942 (95% CI [0.8923–0.9714]) and the CF model had the lowest AUC of 0.846 (95% CI [0.7697–0.9027]) for the five-year survival prediction (Table 4).

The PWE model had significantly better performance than the RAT model for one-year survival prediction (*p* < 0.01, chi-square test). For the three-year and five-year survival prediction, the performance of PWE and RAT models are similar and there was no significant difference (Table 5). Nevertheless, both RAT and PWE had good performance in terms of accuracy. PWE obtained the best accuracy of 0.9107 for three-year survival. Both RAD and PWE performed better than CF with accuracy ranging from 0.8594 (RAT five-year survival) to 0.9107 (PWE, three-year survival) (Table 6).

## 4. Discussion

Our radiomics-clinical model demonstrates the value of combining radiomic features with clinical factors for predicting the prognosis of NSCLC with probability weighting. The model achieved a higher level of predictive accuracy of 0.9107 compared to traditional clinical factors with the highest accuracy of 0.8281 alone, indicating that the combined PWE model can provide valuable information that is not captured by clinical factors alone.

We noted that there were attempts to combine clinical information with radiomics features to predict cancer treatment prognosis such as ridge regression [14,25], logistic regression [26], and Cox regression [24] and obtained an AUC ranging from 0.733 [24] to 0.868 [25,26]. In our model, the probabilistic weighted method takes into consideration that radiomics features and clinical factors are two distinctive factors of different natures and should not be put together as inputs for machine learning. By using a probability-weighted strategy, we obtained a better AUC of 0.955 and an accuracy of 0.9107.

Our study illustrated that prognosis prediction of cancer, in particular NSCLC can be achieved by machine learning models with radiomic features or clinical factors. The advantage of clinical data is the convenience in data collection, such as demographics information of the patients, for example, age and gender. For radiomics prediction, it is a non-invasive method to predict prognosis based on radiomic features extracted from medical images. However, radiomics fail to consider the deterministic factors that significantly influence the prognosis of the patients, which may jeopardize the predictive ability of the model. From our study, it was acknowledged that age was an influential clinical factor affecting the prognosis of the patients. The probability-weighted enhanced model proposed in this study can incorporate clinical data with radiomic features taking into consideration each set of data to achieve a better predictive power than each factor alone.

Radiomics is a promising approach not only in the prognosis of cancer but it can be used for the diagnosis of diseases. By using image analysis techniques, image feature as another form of radiomics can identify subtle differences in tissue properties that may not be visible to the naked eye, and can potentially improve diagnostic accuracy such as automatic detection of ischemic stroke in the brain using CT images [27], lung cancer diagnosis [28,29], prostate cancer detection [30], and brain tumor assessment and classification [31]. In addition, the AI model using radiomics can be applied for histology image classification [32]. Overall, radiomics represent a promising approach for disease diagnosis and prognosis prediction, and with the advancement of this area, radiomics will play an increasingly important role in medical science.

There are factors that may affect the outcomes and the accuracy of the model. These include:

Image quality: types of imaging modality such as CT or ultrasound, image resolution, and image noise can affect the accuracy of the radiomic model [33].

Feature extraction: in this study, feature extraction is based on gross tumor volume delineation. The quality and reproducibility of the radiomic features extracted from the images are dependent on the experience of the oncologists and technologists [34].

Sample size: The size and composition of the dataset used to train and validate the model can impact its accuracy [35].

Treatment effects: When validating a radiomic model, it is important to carefully consider the potential effect of treatment on the accuracy of the model. This may involve including treatment-related variables in the model or stratifying the dataset based on treatment status to ensure that the model is accurate and generalizable to the target population. In our case, we divided the sample into one-, three-, and five-year endpoints [36].

One limitation of our study was the clinical data we collected, such as smoking status and family history, were not included in the data source. This missing information could potentially improve the accuracy of the prediction models.

Another limitation of our study is the lack of clinical validation. Clinical validation is important to confirm the generalizability of our model to other patient populations and healthcare settings. Future studies should aim to validate our model externally using independent datasets.

Despite these limitations, our radiomics-clinical model has important implications for the prognosis of NSCLC patients. The model can provide more accurate and individualized predictions of patient outcomes, which can aid in treatment planning and improve patient survival.

## 5. Conclusions

In this study, we presented a radiomics-clinical probabilistic model for the prognosis of NSCLC. The model combines radiomic features extracted from CT images with clinical factors such as age, histology and tumor stage to predict overall survival. Our results demonstrate the potential of combining radiomics-clinical factors with probability weighting for improving the prognosis of NSCLC patients. Future studies with larger datasets and external validation are needed to confirm the robustness and generalizability of our model.

## Figures and Tables

**Figure 1 biomedicines-11-02093-f001:**
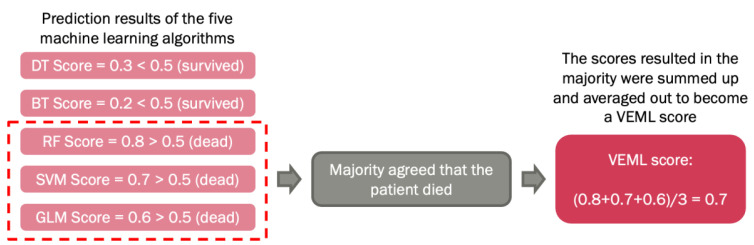
Schematic diagram of voted ensemble machine learning model.

**Figure 2 biomedicines-11-02093-f002:**
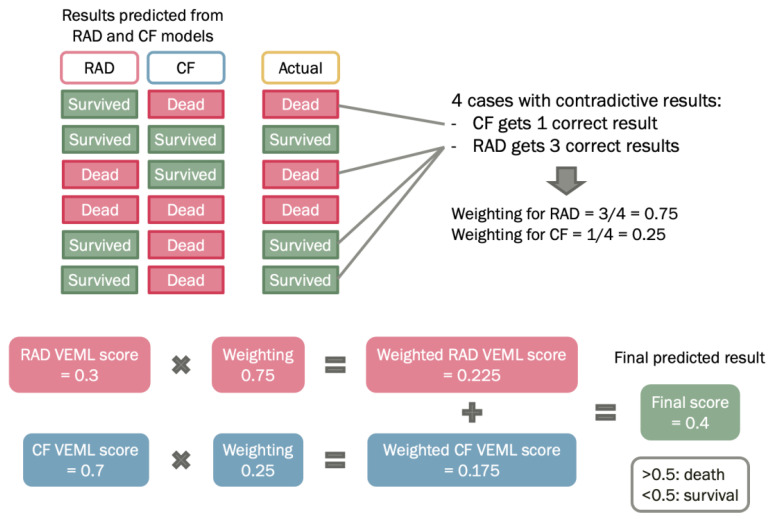
Schematic diagram for the probability-weighted enhanced model (PWEM).

**Figure 3 biomedicines-11-02093-f003:**
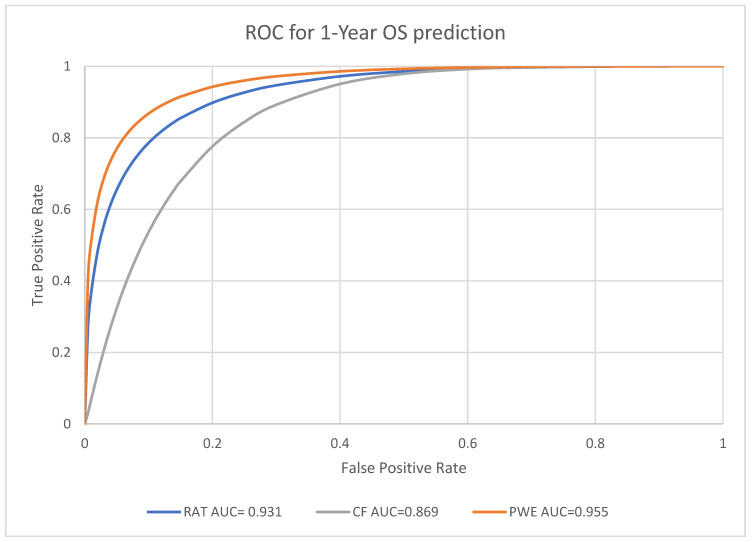
Prediction of 1-year survival using RAT, CF, and PWE models.

**Figure 4 biomedicines-11-02093-f004:**
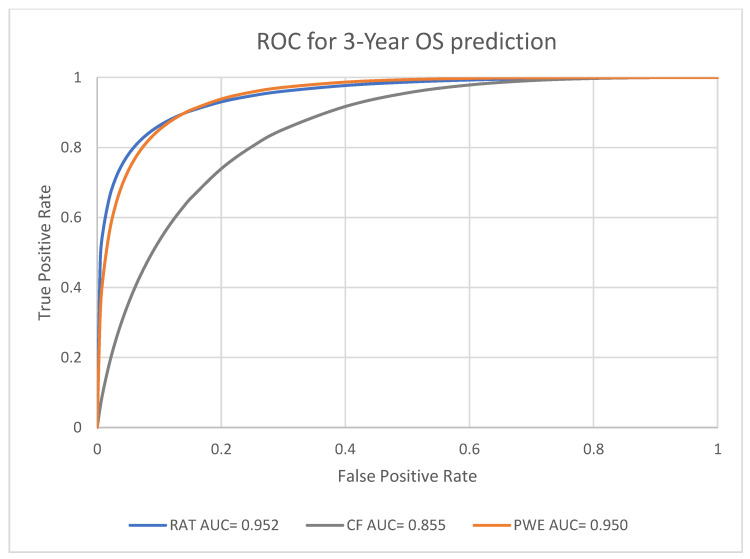
Prediction of 3-year survival using RAT, CF, and PWE models.

**Figure 5 biomedicines-11-02093-f005:**
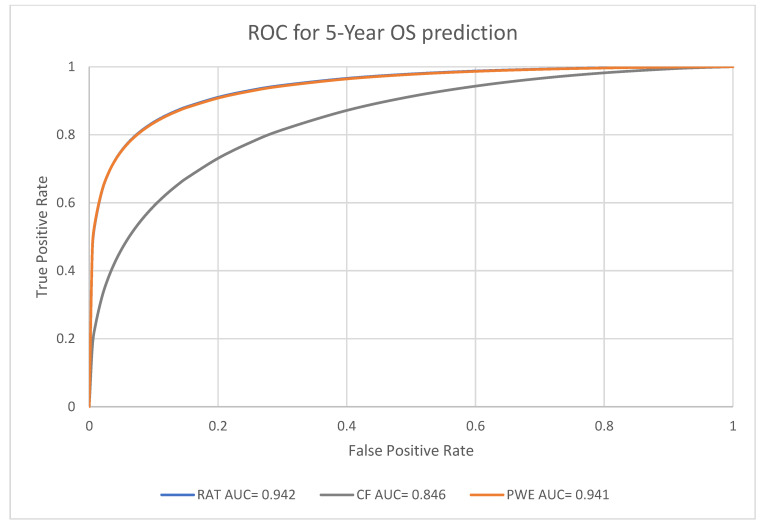
Prediction of 5-year survival using RAT, CF, and PWE models.

**Table 1 biomedicines-11-02093-t001:** Radiomic features summary.

Feature Group	Number of Features
Shape	14
First-order feature	18
Gray level co-occurrence matrix	24
Gray level dependence matrix	14
Gray level run length matrix	16
Gray level size zone matrix	16
Neighborhood gray tone difference matrix	5
*Total*	*107*

**Table 2 biomedicines-11-02093-t002:** Balanced sample size at various endpoints.

Endpoint	1-Year Survival	3-Year Survival	5-Year Survival
		224	Sample size
Balanced sample	119 alive 119 dead	112 alive 112 dead	64 alive 64 dead

**Table 3 biomedicines-11-02093-t003:** Patient demographics and tumor characteristics.

Patient Demographics			
No. of Patients (%)			No. of Patients (%)
Gender		Age	
Male	237 (67%)	≤65 y/o	135 (38%)
Female	115 (33%)	>65 y/o	217 (62%)
Overall Stage		T Stage	
I	60 (17%)	T1	63 (18%)
II	35 (10%)	T2	135 (38%)
IIIa	103 (29%)	T3	49 (14%)
IIIb	154 (44%)	T4	105 (30%)
Histology		N Stage	
Adenocarcinoma	48 (14%)	N0	131 (37%)
Large Cell Carcinoma	105 (30%)	N1	20 (5%)
Squamous Cell Carcinoma	142 (40%)	N2	125 (36%)
Not Otherwise Specified	57 (16%)	N3	73 (21%)
		N4	3 (1%)

**Table 4 biomedicines-11-02093-t004:** Summary of predictive performance of ML models.

Endpoint	Machine Learning Model	AUC [95% Confidence Interval]
	Radiomic model	0.931, [0.894, 0.956]
1-year survival	Clinical factors model	0.869, [0.817, 0.909]
	Probability weighted enhanced model	0.955, [0.926, 0.974]
	Radiomic model	0.952, [0.921, 0.973]
3-year survival	Clinical factors model	0.855, [0.801, 0.898]
	Probability weighted enhanced model	0.950, [0.919, 0.971]
	Radiomic model	0.942, [0.892, 0.971]
5-year survival	Clinical factors model	0.846, [0.770, 0.903]
	Probability weighted enhanced model	0.941, [0.891, 0.971]

**Table 5 biomedicines-11-02093-t005:** Summary of significant difference between models (Chi-square test value and *p* value).

Survival Year(s)	RAT|CF	RAT|PWE	CF|PWE
1	8.0667	10.5986	21.708
	(*p* < 0.05)	(*p* < 0.05)	(*p* < 0.05)
3	18.2596	2.2314	21.9264
	(*p* < 0.05)	(*p* > 0.05)	(*p* < 0.05)
5	10.1110	0.38	17.8133
	(*p* < 0.05)	(*p* > 0.05)	(*p* < 0.05)

**Table 6 biomedicines-11-02093-t006:** Summary of predictive performance of machine learning models in sensitivity, specificity, and accuracy.

Survival Year(s)		RAT	CF	PWE
	1	0.9244	0.9076	0.9244
Sensitivity	3	0.9107	0.8661	0.9196
	5	0.7656	0.7969	0.7813
	1	0.8487	0.6723	0.8487
Specificity	3	0.9018	0.7232	0.9018
	5	0.9531	0.8594	0.9531
	1	0.8866	0.7899	0.8866
Accuracy	3	0.9063	0.7946	0.9107
	5	0.8594	0.8281	0.8672

## Data Availability

Publicly available datasets were analysed in this study. These data can be found at: https://wiki.cancerimagingarchive.net/display/Public/NSCLC-Radiomics (accessed on 22 January 2023). Data used in the preparation of this article were obtained from The Cancer Imaging Archive (TCIA): Maintaining and Operating a Public Information Repository.

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
