# Peer review of "Radiomics-Clinical AI Model with Probability Weighted Strategy for Prognosis Prediction in Non-Small Cell Lung Cancer"

_biomedicines, 2023, doi:10.3390/biomedicines11082093_

Round 1

Reviewer 1 Report

The manuscript submitted for publication to Biomedicines by Tang et al., titled: "Radiomics-clinical AI model with probability weighted strategy for prognosis prediction in non-small cell lung cancer" is an interesting approach towards improving the targeted manner in which treatment for NSCLC can be extended. The reviewer would like to raise the following points for the authors' consideration:

1. Abstract is missing

2. How was the number of observations/cases selected (please provide more information and the rationale used more explicitly)? What were the inclusion and exclusion criteria? How was the number of cases selected (power calculation?).

3. Was there a biochemical validation (counter sample) used for the verification of the machine learning outcomes?

4. It would be interesting to discuss if the obtained results would be similar in terms of their quality if used in other types of cancers. Or even diseases based on the analysis of the tissue damage for instance.

5. The discussion lacks citations. What else has been done in the field? How do the findings of the authors compare and contrast to the literature?

6. What are some effectors/factors that may change the outcomes and the accuracy of the method? Would treatment have an effect (from a validation stand point)

7. The manuscript is significantly under referenced.

English is OK needs proofreading from a native speaker for typos and for flow and optimizing grammar and syntax.

Reviewer 2 Report

The authors examined prognostic measures for non-small cell lung cancer using the Radiomics-clinical AI model. The results showed that the Radiomics model and the clinical model, when used in a probability-weighted strategy, provided better prognostic estimates than the Radiomics model alone.

The study would not be problematic.

But there is no abstract! The authors must write one.

And, for example, how Radiomics is applied, for example, in this case for prognostication, but what about diagnostics? Can't we make a model that predicts whether a patient has lung cancer or not, as well as its histological system? It would be good if you could add such aspects in the discussion.

Round 2

Reviewer 1 Report

The authors have made a reasonable effort to address reviewer's comments. Proofreading is suggested.